# Real-World Evidence: How Long Do Our Patients Fast?—Results from a Prospective JAGO-NOGGO-Multicenter Analysis on Perioperative Fasting in 924 Patients with Malignant and Benign Gynecological Diseases

**DOI:** 10.3390/cancers15041311

**Published:** 2023-02-18

**Authors:** Maximilian Heinz Beck, Derya Balci-Hakimeh, Florian Scheuerecker, Charlotte Wallach, Hannah Lena Güngor, Marlene Lee, Ahmed Farouk Abdel-Kawi, Jacek Glajzer, Jekaterina Vasiljeva, Karol Kubiak, Jens-Uwe Blohmer, Jalid Sehouli, Klaus Pietzner

**Affiliations:** 1Department of Gynecology, Breast Center, Campus Mitte, Charité—Universitätsmedizin Berlin, Corporate Member of Freie Universität Berlin, Humboldt-Universität zu Berlin, Berlin Institute of Health, 10117 Berlin, Germany; 2Young Academy of Gynecologic Oncology (JAGO), Nord-Ostdeutsche Gesellschaft für Gynäkologische Onkologie, 13359 Berlin, Germany; 3Department of Gynecology, St. Joseph Hospital, 12101 Berlin, Germany; 4Department of Gynaecology and Gynaecologic Oncology, University Medical Center Hamburg-Eppendorf, 20251 Hamburg, Germany; 5Department of Gynecology, Katholisches Marienkrankenhaus—Klinik für Gynäkologie, 22087 Hamburg, Germany; 6Department of Gynecology, Center for Oncological Surgery, Campus Virchow Klinikum, Charité-Universitätsmedizin Berlin, Corporate Member of Freie Universität Berlin, Humboldt-Universität zu Berlin, Berlin Institute of Health, 13353 Berlin, Germany; 7Department of Gynecology, Faculty of Medicine, University of Assiut, Assiut 71515, Egypt; 8Department of Gynecology and Obstetrics, Breast Center Ostsachsen, Klinikum Oberlausitzer Bergland Zittau/Ebersbach, 02730 Ebersbach, Germany; 9Department of Gynecology, Vivantes Klinikum Am Urban, 10967 Berlin, Germany; 10Department of Gynecology and Obstetrics, St. Franziskus Hospital Muenster, 48145 Muenster, Germany

**Keywords:** perioperative fasting, ERAS, real-world evidence

## Abstract

**Simple Summary:**

The concept of ERAS (Enhanced Recovery After Surgery) was introduced to reduce perioperative morbidity through a multimodal approach. Optimized and shortened perioperative fasting is a fundamental part of this modern concept of perioperative patient management, as prolonged fasting before and after surgery is associated with unfavorable outcomes. So far, it remains unclear whether increasingly established ERAS protocols lead to adequate short fasting intervals in clinical routines. We therefore conducted this prospective multicenter study and collected real-world data from 924 patients to evaluate actual perioperative fasting behavior. Patients reported drastically prolonged perioperative fasting durations. Even longer fasting intervals were reported for oncological and extensive procedures. Our data suggest that modern optimized fasting management is poorly implemented in clinical routine practice. This study should draw attention to the need for adequate implementation of ERAS protocols and sensitize clinicians to appropriate patient education about perioperative fasting.

**Abstract:**

Background: Despite the key role of optimized fasting in modern perioperative patient management, little current data exist on perioperative fasting intervals in routine clinical practice. Methods: In this multicenter prospective study, the length of pre- and postoperative fasting intervals was assessed with the use of a specifically developed questionnaire. Between 15 January 2021 and 31 May 2022, 924 gynecology patients were included, from 13 German gynecology departments. Results: On average, patients remained fasting for about three times as long as recommended for solid foods (17:02 ± 06:54 h) and about five times as long as recommended for clear fluids (9:21 ± 5:48 h). The average perioperative fasting interval exceeded one day (28:23 ± 14:02 h). Longer fasting intervals were observed before and after oncological or extensive procedures, while shorter preoperative fasting intervals were reported in the participating university hospitals. Smoking, treatment in a non-university hospital, an increased Charlson Comorbidity Index and extensive surgery were significant predictors of longer preoperative fasting from solid foods. In general, prolonged preoperative fasting was tolerated well and quality of patient information was perceived as good. Conclusion: Perioperative fasting intervals were drastically prolonged in this cohort of 924 gynecology patients. Our data indicate the need for better patient education about perioperative fasting.

## 1. Introduction

Preoperative fasting is advised to minimize the risk of pulmonary aspiration of gastric contents during anesthesia [1]. Therefore, recent international guidelines recommend a minimum fasting interval of 6 h for solid foods and 2 h for clear fluids before surgery [1,2]. On the other hand, excessively long fasting before and after surgery should be avoided as prolonged fasting is not only associated with increased patient discomfort [3,4], but also with decreased insulin sensitivity [5,6,7,8], altered inflammatory response [7,8] and higher surgical complication rates [3]. With the introduction of the ERAS (Enhanced Recovery After Surgery) concept, perioperative management has been rethought during the last two decades [9]. The concept of ERAS aims to reduce perioperative morbidity through a multimodal approach by attenuating the patient’s stress response before, during and after surgery, thereby promoting early and complete recovery from surgery [9,10,11,12,13]. A key component of ERAS in this regard is the optimization of perioperative fasting management with the avoidance of excessive fasting and the implementation of an early postoperative restorative diet [9,10,11,12,13]. In contrast to these recommendations, clinical data on perioperative fasting habits has consistently reported drastically prolonged perioperative fasting intervals in different health care systems and settings over the last decades [14,15,16,17,18,19,20,21]. So far, it remains unclear whether increasing implementation of ERAS protocols has led to optimized shorter fasting intervals in clinical routine practice. Therefore, we conducted this study to provide real-world evidence on actual perioperative fasting habits in a multicenter cohort and identify risk factors for excessive fasting. 

## 2. Materials and Methods

### 2.1. Design and Participants

This multicenter cross-sectional study assessed perioperative fasting intervals in a cohort of gynecology patients with the use of a specifically developed questionnaire. The study was conducted in 13 German gynecology departments in 5 different federal states, from 15 January 2021 until 31 May 2022. Included were inpatients after gynecological surgery during their hospital stay. No restrictions were applied regarding indication or type of surgery. Exclusion criteria were age under 18, incapacitated patient or language barriers. Obstetric patients were not included in this analysis. After study inclusion, patients were interviewed by questionnaire. Repeat visits were conducted until the patient’s first postoperative food intake. Prior to inclusion, written informed consent was obtained from all patients after supplying appropriate verbal and written information. The study was primarily approved by the local ethics committee at the Charité university hospital of Berlin. Secondary evaluations and approvals were carried out by the respective responsible ethics committees of the participating study centers. 

The study was developed as part of a scientific and clinical fellowship program of the JAGO—the Young Academy of Gynecologic Oncology (“Junge Akademie Gynäkologische Onkologie”—JAGO) of the Northeastern German Society of Gynecologic Oncology (“Die Nord-Ostdeutsche Gesellschaft für Gynäkologische Onkologie”—NOGGO). 

### 2.2. Questionnaire 

The questionnaire was developed in a five-part modular workshop under advice and with the input of interprofessional and interdisciplinary experts. The questionnaire was divided into two parts. The first part was completed by the patient, and the second part by the respective local study investigator. The questionnaire for the patients consisted of 22 items. Patients documented the time of their last preoperative food and fluid intake and, if applicable, the time they last smoked. The time of the patient’s first postoperative food intake and the postoperative use of high calorie drinks were also recorded. In addition, the questionnaire included questions about the patient’s condition during fasting, experience with fasting, the quality of patient education about fasting and subjective perceptions of fasting. While most questions were dichotomous, with one answer option, the items on medical information and subjective beliefs about fasting had the possibility of multiple answers. Questions on condition during fasting were documented via graduated scales. The quality of medical information on fasting was measured via a rating scale. The goal of the questionnaire was to reflect management regarding preoperative fasting from the patient’s perspective. The questionnaire was tested in a run-in phase in ten patients to test understanding and readability. 

The second part consisted of the following clinical characteristics and demographic data, that were assessed by the respective study investigators: age, height, weight, surgical procedure, indication for and length of surgery, day of hospital admission, urgency of surgery, chronic disease, medication, American Society of Anesthesiologists Physical Status Classification and Charlson Comorbidity Index. 

As the study included a broad spectrum of surgeries, a four-level classification system for the extent of surgery has been designed and applied, ranging from small to moderate, complex and extensive surgery. Small procedures were defined as short open or endoscopic procedures such as hysteroscopies or conizations; moderate procedures were defined as uncomplicated open or endoscopic procedures such as breast conserving surgeries, non-complex laparoscopies or uncomplicated urogynecologic procedures; complex procedures were defined as advanced open or endoscopic procedures such as resection of extensive endometriosis, complex breast reconstruction, complex myomectomy or hysterectomy; extensive procedures were defined as extensive open or endoscopic procedures such as advanced oncologic surgeries, debulking surgery, radical hysterectomy or free flap transplantation. The classification was made by the respective local study investigator. 

### 2.3. Data Analysis

Descriptive statistics were used to report results of the questionnaire. Unless stated otherwise, continuous variables are presented as mean and standard deviation. The duration of fasting is presented in hours and minutes (i.e., hh:min) and categorical variables are presented as percentage and number (i.e., % (*n*)). Preoperative fasting intervals were calculated as the time difference between documented last preoperative food or fluid food intake and the beginning of surgery, defined as incision time. Postoperative fasting duration was calculated as the difference between the end of surgery, defined as suture time, and documented first postoperative food intake. The total perioperative fasting interval was defined as the time span between documented last preoperative and first postoperative food consumption. 

To further evaluate which patients were at risk for prolonged fasting from solid foods before surgery, we predefined the following three subgroups of preoperative fasting intervals: (1). patients who fasted less for than 10 h (adequate), (2). patients who fasted for 10–18 h (prolonged) and (3). patients who fasted for more than 18 h (excessive). Baseline characteristics were compared between the respective subgroups. Further subgroup analyses of fasting intervals were performed for indication of surgery (benign/malignant), extent of surgery, abdominal manipulation (extra- or intraperitoneal surgery) and type of hospital (university/non-university). Group differences were examined using unpaired two-tailed t-tests and ANOVA. A multiple linear regression model was applied to test whether clinical characteristics can predict preoperative fasting intervals. Predefined predictor variables were age, body mass index, smoking, indication of surgery (benign or malignant), extent of surgery category, day of hospital admission and treatment in a university or non-university clinic. The significance level of all tests was defined to be alpha <0.05 (* <0.05, ** < 0.01, *** < 0.001, **** <0.0001). Study data were collected and managed using REDCap ^®®^ software [22]. The statistical analysis was performed using SPSS Version 28 ^®®^ (IBM Corp., Armonk, NY, USA) and Prism Version 9 ^®®^ (GraphPad Software Inc., San Diego, CA, USA).

## 3. Results

### 3.1. Baseline Characteristics

Overall, 924 female patients with an average age of 52 ± 15.2 years were included in this study, from 13 different gynecology departments. The participating study sites included three university departments and 10 non-university departments. Of these patients, 56.2% (519) underwent surgery for benign disease and 43.8% (405) for malignant disease. The vast majority (92.8% (857)) were elective surgeries and average surgery time was 107 ± 82.3 min. As described previously, we introduced a four-grade classification system to categorize the extent of surgery. About half of the procedures were categorized as moderate (51.1% (472)), followed by complex (27.7% (256)), extensive (14.9% (138)) and small (6.3% (58)) surgeries. As breast diseases are treated by gynecologists in Germany, both gynecologic (74.5% (688)) and senologic (25.5% (236)) surgeries were represented in this study. A more detailed overview of patient characteristics and surgical procedures and indications is provided in Table 1. 

Incomplete data were noted in seven patients for preoperative fasting from solid foods and in 13 patients for preoperative fasting from fluids. For 78 patients, the first postoperative food intake was not documented. Complete data on perioperative fasting intervals were documented in 831 patients. No patients were excluded from the analysis.

### 3.2. Duration of Perioperative Fasting 

#### 3.2.1. Perioperative Fasting from Solid Foods 

The average duration of preoperative fasting from solid foods was 17:02 ± 06:54 h, Figure 1 and Table 2. Only 4.1% (38) of patients showed adequate adherence to fasting recommendations, with a preoperative fasting duration of less than 10 h for solid meals, and almost one third (31.3% (289)) of patients fasted for longer than 18 h before surgery, Table 3. All participants complied to the required fasting duration of 6 h for solid foods (minimum fasting duration 6:35 h). The first postoperative intake of solid food was documented after an average of 9:42 ± 12:00 h and the total average perioperative fasting interval for solid food was 28:23 ± 14:02 h, Figure 1.

We also documented whether patients received high calorie drinks for early postoperative nutrition. Only 5.8% (54) of the participants stated that they received high calorie drinks postoperatively. 

#### 3.2.2. Preoperative Fasting from Fluids

The average preoperative fasting interval for fluids was 9:21 ± 5:48 h, Figure 1. Only a minority of patients (18.9% (175)) showed adequate adherence to fasting recommendations, with a fasting duration of less than 4 h for fluids. A significant proportion of patients (42% (388)) did not consume fluids for more than 10 h preoperatively. Seventeen participants did not comply to the required minimum 2 h of no preoperative fluid intake (minimum fasting duration for fluids 26 min). 

#### 3.2.3. Preoperative Deviation from Intake Habits

Smokers showed a large variance regarding preoperative nicotine abstinence, with a median 13:02 h (IQR 06:18—17:57 h) of non-smoking. Of the participants, 47.3% (437) reported taking medication at fixed times. Of those, 41.1% (180) deviated from fixed intake times during preoperative preparation.

### 3.3. Risk Factors for Prolonged Fasting 

#### 3.3.1. Oncologic Surgery

Patients who had surgery for malignancies fasted significantly longer from solid foods before and after surgery, compared to patients with surgery for benign indications (pre-op: Δ 60.51 min; post-op: Δ 161.6 min), Table 2. Subsequently, the total perioperative fasting intervals for solid foods were also prolonged for patients undergoing oncologic surgery (Δ 244.7 min, *p* < 0.0001). Preoperative fluid restriction did not differ between benign and malignant surgical indications. 

#### 3.3.2. Abdominal Manipulation 

Patients who had intraperitoneal surgery (laparoscopy, laparotomy) showed clearly longer postoperative fasting intervals for solid foods, compared to patients who did not have abdominal manipulation during surgery (Δ 336.1 min), Table 2. Subsequently, total perioperative fasting intervals for solid foods were also prolonged for patients who had abdominal manipulation during surgery (Δ 396.8 min, *p* ≤ 0.001). Preoperative fasting intervals for fluids and solid foods did not differ between intra- and extraperitoneal procedures. 

#### 3.3.3. Extent of Surgery

Patients who had extensive surgery reported significantly prolonged preoperative fasting intervals from solid foods, compared to patients who had moderate (Δ 130.6 min) or complex (Δ 127.7 min) surgery. However, no difference was found between small and extensive procedures, Table 2. Longer postoperative and consecutively prolonged overall perioperative fasting intervals could be observed with increasing extent of the surgery category, Table 2. Preoperative fluid restriction did not differ between the extent of surgery categories in the multiple comparison analysis, Table 2.

#### 3.3.4. Type of Hospital 

On average, patients treated in a university hospital showed relevant shorter preoperative fasting intervals both for fluids (Δ 97.1 min) and solid foods (Δ 66.2 min), compared to patients from non-university hospitals, Table 2. In contrast, the duration until the first postoperative solid food intake was significantly longer for patients treated at university hospitals (Δ 117.6 min). Total perioperative fasting intervals from solid foods did not differ between university and non-university hospitals.

#### 3.3.5. Patient Characteristics of Preoperative Fasting Categories

To understand which patients might comply with fasting recommendations and which patients are at risk for prolonged fasting, a comparison of baseline characteristics was made for the predefined subgroups of preoperative fasting durations (fasting interval for solid foods <10 h, 10–18 h, >18 h). In summary, patients who fasted for less than 10 h were significantly younger, had fewer comorbidities resulting in lower ASA scores and lower Charlson Comorbidity Indices, were more frequently operated on for benign conditions and underwent less extensive surgeries with lower extents of surgery categories, compared to patients with longer fasting intervals. Patients who fasted for longer than 18 h were more often hospitalized before the day of surgery and were more often operated on for gynecologic disease, compared to patients with fasting durations between 10 and 18 h. No differences were found for the type of hospital, duration of surgery, history of smoking, chronic medication or patients’ BMI between the predefined subgroups. For a detailed overview of the subgroup analysis see Table 3.

A multiple linear regression model was applied to test whether age, BMI, smoking, non-age adjusted Charlson Comorbidity Index, time of hospital admission, type of hospital, indication for surgery (benign vs. malignant) or extensive surgery could predict preoperative fasting durations, both for solid foods and fluids. The overall regression was statistically significant, for both models.

The fitted regression model for preoperative fasting from solid foods was statistically significant (R2 = 0.5, F (8, 866) = 5743, *p* <0.001). It was found that treatment in a non-university hospital (β = 110.7 min, *p* < 0.001), increasing score of the Charlson Comorbidity Index (β = 25.8 min, *p* = 0.008), extensive surgery (β = 61.6 min, *p* = 0.043) and smoking (β = 61.6 min, *p* = 0.015) significantly predicted longer preoperative fasting durations from solid foods. Age, BMI, surgical indication and time of hospital admission were not significant predictors of the applied model. 

The fitted regression model for preoperative fasting from fluids was also statistically significant (R2 = 0.28, F (8, 860) = 3128, *p* = 0.002). It was found that treatment in a non-university hospital (β = 95.5 min, *p* < 0.001) and hospitalization before the day of surgery (β = 54.2 min, *p* = 0.03) significantly predicted longer preoperative fasting durations from fluids. In contrast, treatment for malignant disease predicted a shorter fasting duration from fluids (β = −70.0 min, *p* = 0.036). Age, BMI, Charlson Comorbidity Index, smoking and extensive surgery were not significant predictors of the applied model. 

### 3.4. Condition during Fasting 

In general, prolonged fasting intervals were tolerated well by the majority of patients, Figure 2. Most patients stated they had no or only slight difficulties in complying with fasting recommendations. About two-thirds of patients documented no or slight thirst and few patients reported moderate or strong thirst. Most said they did not feel hungry, and the majority did not feel weak during preoperative fasting. 

### 3.5. Information about Fasting 

Patients were asked who informed them about fasting recommendations, and had the option of multiple answers. Most frequently, patients were advised on preoperative fasting by anesthesiologists 87.7% (810), and only in 26.1% (241) of cases, the surgeon/gynecologist informed patients on fasting. Nursing staff informed a further 44.2% (408) of patients. The general quality of information was perceived as good; on average patient education was rated 8.15 ± 2.37 on a rating scale from 0 (worst) to 10 (optimal). 

Participants were also asked what would have helped them to comply with fasting recommendations, and had the option of multiple answers. The most frequently selected answer was “no further support needed” (43.1% (398)), followed by “more detailed information from the anesthesiologist” (24.5% (226)) and “more information about reasons for fasting recommendations” (23.6% (218)). The options “information brochure (15.7% 145))”, “more detailed information from the surgeon” (8.7% (80)), “app-based information” (2.1% (19)) and “podcast-based information” (0.9% (8)) were less frequently selected. 

To evaluate the quality of information on fasting recommendations, patients were asked what may be ingested two hours before surgery, and had the option of multiple answers. The majority (90.4% (835)) selected the correct answer, “water”, but only a few selected the other correct answer, “clear apple juice” (8% (74)). The majority recognized the incorrect answers as “orange juice” (1.4% (13)), “banana” (2.8% (26)), “milky coffee” (1.8% (17)), and “cigarette” (2.7% (25)). When asked for the reason for preoperative fasting, most patients named correctly the “risk reduction for pulmonary aspiration of gastric contents” (75.6% (699), followed by “risk reduction for postoperative complications” (40.6% (375)), “risk reduction for infections (11.4% (105)), “risk reduction for bleeding” (10.1% (93)) and “better wound healing” (8.9% (82)). Most patients believed that adherence to fasting recommendations would improve the outcome of surgery (“yes” 68.0% (628), “unsure” 21.6% (200), “no” 8.1% (75)).

## 4. Discussion

### 4.1. Summary of Results

This multicenter cross-sectional study clearly demonstrates poor adherence to guideline fasting recommendations and drastically prolonged perioperative fasting intervals in a cohort of gynecology patients. The average preoperative fasting duration was 17:02 h for solid foods and 09:21 h for fluids. Only a small minority of patients adhered to fasting recommendations, with a preoperative fasting interval of less than 10 h for foods and less than 4 h for fluids. Nevertheless, most patients tolerated the prolonged preoperative fasting well. In general, the quality of information on fasting recommendations was perceived as good. 

To our knowledge, we are the first to identify risk factors for prolonged preoperative fasting in a gynecological cohort. Patients who underwent extensive surgery and oncologic surgery had significantly prolonged pre- and postoperative fasting intervals, whereas shorter preoperative fasting intervals were observed in the participating university hospitals. Patients who underwent abdominal manipulation showed longer post- and perioperative fasting durations. Patients who adhered to fasting recommendations were generally younger, had fewer comorbidities, underwent less extensive surgery, were operated on more often for benign diseases, and were usually hospitalized on the day of surgery. The applied linear regression model revealed that smoking, treatment in a non-university hospital, increasing comorbidities and extensive surgery were significant predictors for prolonged preoperative fasting from solid foods, while treatment in a non-university hospital and hospitalization prior to surgery predicted longer preoperative fasting durations for fluids. 

### 4.2. Comparison with the Literature 

Findings from this large multicenter study are consistent with previously published data over the past decades, showing prolonged perioperative fasting intervals in different health care systems and settings. In comparison to mostly uni- or bicentric cohort studies with smaller sample sizes, our study tended to show even longer fasting intervals [14,15,16,17,18,19,20,21]. Breuer and colleagues (2010) published a monocentric survey on perioperative fasting intervals at the in-house department of anesthesiology of the Charité university hospital, with similar results [19]. Twelve years later, we could not observe any relevant improvements in preoperative fasting habits in our cohort under comparable local conditions. The average preoperative food restriction was, in fact, about 2 h longer, compared to the reported data from Breuer and colleagues [19]. Another recently published Dutch study also found no relevant changes in perioperative fasting durations over the course of 10 years [18]. These data clearly demonstrate the difficulties of change management. Further studies should define the various barriers and should investigate possible actions to overcome these challenges. 

Our findings contrast with current ERAS recommendations [11,12,23]. As prolonged preoperative fasting is related to patient discomfort [3,4], reduced insulin sensitivity [5,6,7,8], altered inflammatory response [7,8] and higher surgical complication rates [3], shortened perioperative fasting intervals are recommended in ERAS protocols [11,12,23]. ERAS recommendations to minimize perioperative fasting intervals include preoperative consumption of carbohydrate enriched fluids up to two hours before surgery (“carbo-loading”) [24,25,26,27] and early restorative diet after surgery, with the use of high calorie drinks [12]. These recommendations are not reflected in clinical routines, as our data show. In addition to prolonged fasting intervals and suboptimal postoperative food restoration, only 5% of our participants received high calorie beverages as part of an intensified restorative diet after surgery. Some hope is given by the fact that shorter fasting intervals were reported in the participating university hospitals, suggesting that optimized fasting management might be implemented, at least to some extent. On the contrary, shorter postoperative fasting durations were reported in the participating non-university hospitals, questioning the hypothesis of optimized fasting management in university institutions.

### 4.3. Risk Factors for Prolonged Fasting 

We identified significant predictors for prolonged fasting intervals. Smoking, comorbidities (measured with the Charlson Comorbidity Index), treatment in a non-university hospital and increasing extent of surgery predicted longer preoperative fasting for food. In addition, longer pre- and postoperative fasting intervals were observed in patients who underwent oncological or extensive procedures. This seems contradictory, as one would expect better implementation of ERAS recommendations and thus optimized fasting intervals in the management of oncologic or frail patients undergoing extensive surgery, given the increased procedure- and condition-related morbidity [28,29,30]. Delayed postoperative restoration of oral intake after oncologic and abdominal surgery has been reported previously [18], which can be explained by the increased procedure-related morbidity and procedure-related abdominal manipulation. Nevertheless, an early restorative diet might also be crucial for this cohort of patients with increased perioperative morbidity. In contrast, patients who followed fasting recommendations had a lower risk for perioperative complications as they were significantly younger, had fewer comorbidities, underwent less extensive surgery and were treated more often for benign disease. To summarize, patients with an already increased risk of perioperative complications, due to the type of procedure or comorbidities, appear to fast for even longer. 

### 4.4. Information Needs

Information on the importance of minimum fasting intervals seems to be adequate, as almost all participants complied with required minimum fasting recommendations. Patients mostly felt well informed and were able to cite the risk of pulmonary aspiration as the main reason for preoperative fasting. Nevertheless, there is a lack of awareness about excessively long fasting intervals, as only a few patients had adequately short fasting intervals and the majority fasted for significantly too long. The obsolete recommendation ‘nil by mouth from midnight’ [1] is apparently still present in clinical practice. Adequate patient information on perioperative fasting should therefore include the recommendation to avoid prolonged preoperative fasting intervals and emphasize food and fluid intake close to the recommended minimum fasting limits. As several studies show, implementation of structured patient education is feasible and leads to better adherence to guideline recommendations and shortened preoperative fasting intervals [31,32]. Our patients asked more often for better quality of information on fasting than for potential informational tools, such as brochures or digital media support. In conclusion, detailed patient education on fasting should be a crucial tool to optimize perioperative fasting intervals.

In our study, recommendations on perioperative fasting were mainly provided by anesthesiologists and only rarely by gynecologic surgeons. The underrepresentation of surgeons is surprising, as they are the central perioperative care provider and should therefore play a key role in perioperative patient and fasting management. Leaving patient education on perioperative fasting to a single discipline, namely the anesthesiologist, obviously does not meet the challenges of adequate implementation of ERAS protocols in clinical routine. Patient education and fasting management should, instead, be provided by an interdisciplinary team in which the gynecologic surgeons play a central role. Therefore, awareness on short preoperative fasting intervals and early postoperative nutrition should be raised among gynecologic surgeons. 

### 4.5. Limitations and Strengths 

The present study has certain limitations. First, information on preoperative fasting intervals is based on recollection after surgery and may therefore be inaccurate. In addition, preoperative fasting durations were calculated as the temporal difference between documented last preoperative food or fluid food intake and the beginning of surgery, defined as incision time. This disregards the earlier onset of anesthesiologic measures and could lead to overestimated preoperative fasting intervals. The authors deliberately chose the beginning of surgery as a reference, as the time of incision was considered the most accurately documented time. In our experience, the onset of anesthesiologic measures is usually less accurately recorded and defined. A possible overestimation can be overlooked, given the reported length of perioperative fasting intervals. Nevertheless, preoperative anesthesiologic and surgical preparations might take longer before extensive surgeries. This could be a partial explanation for the observed longer fasting durations before extensive procedures. 

This analysis does not consider the perioperative volume management and a possible associated impact on perioperative fasting intervals. No data were collected on perioperatively administered intravenous infusions or parenteral nutrition. Both could influence pre- and postoperative fasting intervals in various ways. In addition, no data were collected on the use of preoperative carbohydrate enriched fluids before surgery (“carbo-loading”).

Non-German speaking patients are underrepresented in this study as the questionnaire used was only available in German. The questionnaire could be used for a structured interview if the patient and study investigator spoke the same language, but patients with greater language barriers could not be included. Migrants might be even at greater risk for inappropriate perioperative fasting information. Further studies are needed to include patients from different ethnic and cultural backgrounds. 

We introduced a classification for the extent of surgery. The authors are aware that the scoring is partly based on a subjective assessment by the respective local study investigator. The score has also not been independently verified. The system provides an easy-to-use tool to assess the extent of surgery as, to the best of our knowledge, suitable alternatives are not available. Furthermore, the classification is supported by the fact that the duration of surgery increased with higher extent of surgery categories.

We evaluated the patient’s perspective and knowledge but not that of the medical staff. Further trials should investigate staff awareness of guideline fasting recommendations, since knowledge and awareness regarding guidelines and evidence-based procedures seem to be generally lacking [9].

One of the strengths of the present analysis is the multicenter study design. The high proportion of non-university hospitals provides a realistic image of actual clinical routine and practice, especially since patients treated in non-university hospitals are often underrepresented in clinical studies. The relatively large and homogenous cohort gave the possibility for subgroup analyses. Thereby, we were able to identify the above-mentioned risk factors for prolonged fasting. Previous studies were probably unable to identify risk constellations for prolonged preoperative fasting because they comprised smaller and more heterogeneous groups [14,15,16,17,18,19,20,21]. 

## 5. Conclusions

Despite the increasing importance of ERAS protocols, optimized perioperative fasting management is hardly implemented in clinical routine practice and perioperative fasting intervals are still drastically prolonged. Patients at increased risk for perioperative morbidity, due to type of procedure or comorbidities, seem to fast even longer. Better and more detailed information on fasting is needed and should be the main tool for future improvement. It appears that gynecologic surgeons are underrepresented in patient education on perioperative fasting, and they should in future play a greater role in fasting management. Therefore, systematic educational programs for medical staff and regular audits should be implemented to increase adherence to preoperative fasting recommendations and to improve patients’ outcomes.

## Figures and Tables

**Figure 1 cancers-15-01311-f001:**
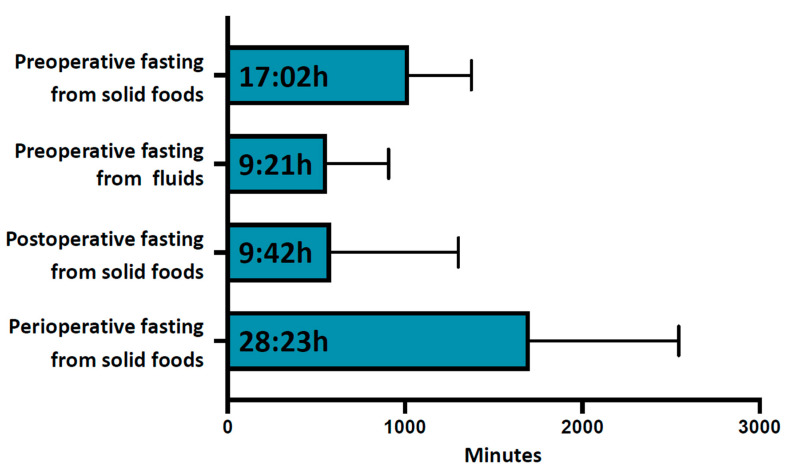
Perioperative fasting intervals. From top to bottom are shown average preoperative fasting duration for solid foods, average preoperative fasting duration for fluids, average postoperative fasting duration for solid foods and average total perioperative fasting duration for solid foods. Data are shown in hh:min. Error bars indicate standard deviation.

**Figure 2 cancers-15-01311-f002:**
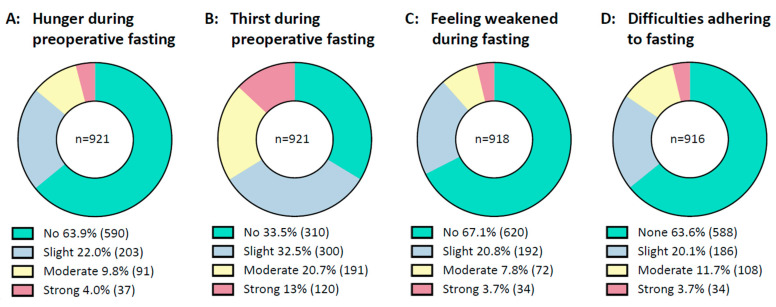
Condition during fasting. From right to left are shown: (**A**): Patients’ feelings of hunger during preoperative fasting. (**B**): Patients’ feelings of thirst during preoperative fasting. (**C**): Patients’ feelings of weakness during preoperative fasting. (**D**): Patients’ difficulties in adhering to fasting recommendations. Data are presented as a circle chart. Green indicates “no or none”, blue indicates “slight”, yellow indicates “moderate” and red indicates “strong”. Bellow the chart, the percentage of answers is displayed next to the legends. The number of patients is displayed in parentheses behind the percentage. *n* = number.

**Table 1 cancers-15-01311-t001:** General characteristics. Patient and procedure related characteristics are displayed. On the righthand side, indications of performed procedures are further specified. SD = Standard deviation, *n* = number.

Age—Mean ± SD (Years)	52 ± 15.2	Indication for Surgery—*n* (%)	924 (100)
**BMI**—mean ± SD	26.4 ± 6.0	Malignant	405 (43.8)
**Chronic medication**—*n* (%)	445 (48.2)	Benign	519 (56.2)
**Chronic disease**—*n* (%)	568 (61.5)	Intraperitoneal procedure	593 (64.2)
**Smoking**—*n* (%)	180 (19.6)	Non-abdominal procedure	331 (35.8)
**Charlson Comorbidity Index**—mean ± SD	1.4 ± 1.8	Breast cancer	197 (21.3)
**ASA Index**—n (%)		Ovarian cancer	100 (10.8)
*ASA-I*	265 (28.7)	Uterine cancer	58 (6.3)
*ASA-II*	518 (56.1)	Cervical cancer	27 (2.9)
*ASA-III*	138 (14.9)	Vulvar/Vaginal cancer	23 (2.5)
*ASA-IV*	3 (0.3)		
**Hospital care level**—n (%)		Benign adnexal tumor	121 (13.1)
University Hospital	325 (35.2)	Uterine fibroma	112 (12.1)
Non-University Hospital	599 (64.8)	Pelvic prolapse and incontinence	46 (5)
**Hospital admission**—n (%)		Endometriosis	42 (4.5)
Day of surgery	688 (75.5)	Benign breast tumor	28 (3.0)
1 Day before surgery	188 (20.3)	Abortion and Extrauterine gravity	27 (2.9)
2 ≤ Days before surgery	35 (3.8)	Irregular bleeding	26 (2.8)
**Extent of surgery**—n (%)		Cervical precancerosis	24 (2.6)
Category I (small)	58 (6.3)	Acute abdominal pain	19 (2.1)
Category II (moderate)	472 (51.1)	Benign endometrial tumor	17 (1.8)
Category III (complex)	256 (27.7)	Genetic high-risk constellation	15 (1.6)
Category IV (extensive)	138 (14.9)	Infertility	12 (1.3)
**Duration of surgery**—mean ± SD [minutes]	107 ± 82.3	Benign vulvar/vaginal disease	14 (1.5)
**Urgency of surgery**—n (%)		Pelvic inflammatory disease	8 (0.9)
Elective	857 (92.8)	Other	8 (0.9)
Urgent	48 (5.2)		
Immediate	18 (1.9)		

**Table 2 cancers-15-01311-t002:** Subgroup analysis for perioperative fasting intervals. From top to bottom are shown overall results and subgroup fasting intervals for hospital type, indication of surgery and procedure category. Data are presented as mean ± standard deviation in minutes. The number (*n*) of patients is displayed in parentheses behind the data. *p*-values of group comparisons are shown in italics: *t*-testing for hospital type and indication for surgery and ANOVA testing for procedure category. Details of multiple comparison testing of the procedure categories, *p*-values are indicated as follows: <0.05, ** < 0.01, *** < 0.001, **** < 0.0001. ^†^ II vs. IV ***, III vs. IV **; ^‡^ No significant results of multiple comparison analysis; ^§^ I vs. III **, I vs. IV ****, II vs. III **, II vs. IV ****, III vs. IV ****.

	Preoperative Fasting Interval from Solid Foods (*n*)	Preoperative Fasting Interval from Fluids (*n*)	Postoperative Fasting Interval from Solid Foods (*n*)
**Overall**	1021.6 ± 353.9 (917)	560.6 ± 347.5 (911)	582.1 ± 720.0 (846)
**Hospital type**			
University	978.8 ± 281.8 (324)	498.2 ± 325.2 (325)	656.4 ± 722.5 (312)
Non-University	1045.0 ± 385.9 (593)	595.2 ± 354.9 (586)	538.7 ± 715.6 (534)
	***p* = 0.0067**	***p* ≤ 0.0001**	***p* = 0.0218**
**Indication for surgery**			
Benign	994.9 ± 306.6 (512)	580.6 ± 360.4 (509)	511.3 ± 612.3 (475)
Malignant	1055.0 ± 403.8 (405)	535.2 ± 329.3 (402)	672.9 ± 830.0 (371)
	***p* = 0.0101**	*p = 0.0503*	***p* = 0.0012**
**Abdominal manipulation**			
Intraperitoneal surgery	1009.5 ± 324.0 (330)	550.7 ± 346.1 (327)	369.2 ± 302.3 (310)
Non-Abdoninal surgery	1028.4 ± 369.3 (587)	566.1 ± 348.5 (584)	705.3 ± 851.2 (536)
	*p = 0.437*	*p = 0.522*	***p* ≤ 0.001**
**Procedure category**			
Category I (small)	1071.1 ± 376.7 (56)	648.2 ± 509.8 (56)	255.1 ± 204.7 (53)
Category II (moderate)	996.5 ± 304.4 (469)	578.8 ± 340.9 (466)	420.1 ± 326.0 (439)
Category III (complex)	999.5 ± 292.6 (254)	524.6 ± 371.2 (251)	612.1 ± 815.7 (236)
Category IV (extensive)	1127.1 ± 538.7 (138)	528.7 ± 347.6 (138)	1273.1 ± 1166.4 (118)
	***p* = 0.0008 ^†^**	***p = 0.0325* ^‡^**	***p* ≤ 0.0001 ^§^**

**Table 3 cancers-15-01311-t003:** Patient characteristics in relation to preoperative fasting categories for solid foods. A = Adequate fasting with a fasting interval of less than 10 h. B = Prolonged fasting with a fasting interval between 10 and 18 h. C = Extensively prolonged fasting with a fasting interval over 18 h. *p*-values are displayed for intergroup differences from ANOVA testing. Significance level with post hoc testing is indicated as follows: * *p* < 0.5, ** *p* < 0.01, *** *p* < 0.001. SD = standard deviation, n = number.

	A: Under 10 h	B: 10–18 h	C: Over 18 h	*p*-Value
**Number**	38	590	289	
**Age**—mean ± SD (years)	43.2 ± 17.1	52.0 ± 14.6	53.1 ± 16.0	**<0.001 (*A-B*** **, ***A-C*** *****)**
**Charlson Comorbidity Index**—mean ± SD	0.5 ± 1.0	1.4 ± 1.7	1.5 ± 1.9	**0.004 (*A-B*** **, ***A-C*** ****)**
**ASA Score**—mean ± SD	1.6 ± 0.6	1.9 ± 0.7	1.9 ± 0.6	**0.022 (*A-B*** *, ***A-C*** ***)**
**Hospital admission**—n (%)				**0.011 (*B-C*** ***)**
Day of surgery	32 (84.2)	454 (78.0)	198 (69.7)	
1 Day before surgery	6 (15.8)	109 (18.7)	72 (25.4)	
2 ≤ Days before surgery	0	19 (3.3)	14 (4.9)	
**Procedure category**—n (%)				**0.004 (*A-B*** *****, A-C*** **)
Category I (small)	5 (13.2)	28 (4.7)	24 (8.3)	
Category II (moderate)	26 (68.4)	308 (52.2)	134 (46.4)	
Category III (complex)	6 (15.8)	166 (28.1)	82 (28.4)	
Category IV (extensive)	1 (2.6)	88 (14.9)	49 (17.0)	
**Discipline**—n (%)				**<0.001 (*B-C*** *****)**
Gynecology	31 (81.6)	407 (69.0)	243 (84.1)	
Senology	7 (18.4)	183 (31.0)	46 (15.9)	
**Indication for surgery**—n (%)				**<0.001 (*A-B*** ******, A-C*** ****)**
Benign	33 (86.8)	313 (53.1)	166 (57.4)	
Malignant	5 (13.2)	277 (46.9)	123 (42.6)	

## Data Availability

The data presented in this study are available on request from the corresponding author.

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
