# Peer review of "Real-World Evidence: How Long Do Our Patients Fast?—Results from a Prospective JAGO-NOGGO-Multicenter Analysis on Perioperative Fasting in 924 Patients with Malignant and Benign Gynecological Diseases"

_cancers, 2023, doi:10.3390/cancers15041311_

Round 1
Reviewer 1 Report
The authors present an interesting study regarding perioperative fasting. The topic seems to be very important in terms of postoperative recovery
However, I have some concerns:
1. There are some language errors in the manuscript. Authors must edit the paper to remove errors.
2. The style of presenting the results (tables and graphs) can be improved to make them more readable for the recipient
Reviewer 2 Report
Congratulations to the authors for tackling such an important topic as perioperative fasting. Research in this area may contribute to the further development of perioperative care standards in the practical aspect.
If possible, I suggest improving the manuscript with the following information:
Results
1. How many of these centers (hospitals) had an implemented and actually functioning ERAS?
2. Were there differences between the fasting time recommended by the hospital staff and the actual fasting time in patients (in each option, i.e. preoperative fasting from solid foods, preoperative fasting from liquids, perioperative fasting)?
3. In the "fasting from fluids" option, I propose to distinguish a group of patients who, after starting "preoperative fasting from solid food", intake carbohydrate drinks (e.g. preOp, apple juice, etc.) and a group that took water or another clear liquid without added carbohydrates .
4. In table 2, please indicate in parentheses the number of patients (n) next to each option in the first column, i.e. "University", "non-university", "malignant", etc., "category I", etc. It will be more convenient for the reader .
5. Did the patients receive intravenous infusions of glucose solution in the perioperative period? This may have influenced the feeling of fasting.
6. What type of anesthesia was used in patients and did it affect the time of postoperative and thus perioperative fasting?
7. How many patients felt nausea and vomiting after the surgery and did it affect the time of postoperative and perioperative fasting?
8. Line 368 "only 5% of our participants received high-calorie drinks as part of an intensified restorative diet after surgery" - Where is this data in the results? Also, please include in the „Results” data on the consumption of carbohydrate drinks in the preoperative period.
Reviewer 3 Report
This manuscript describes the status of the perioperative fasting interval in gynecologic surgery patients. I have a few comments for the author.
Patients with background diseases such as breast cancer and dormitory mammary disease patients are also included. Because these patients do not undergo abdominal surgical manipulation, it is not appropriate to include them in the analysis of the perioperative fasting interval together with other gynecologic surgeries involving abdominal manipulation.
Similarly, with regard to the classification of extent of surgery, we believe that it is more appropriate to distinguish between intraperitoneal operations and other cases if the perioperative fasting interval is to be analyzed.
